# Prevention of P2 Receptor-Dependent Thrombocyte Activation by Pore-Forming Bacterial Toxins Improves Outcome in A Murine Model of Urosepsis

**DOI:** 10.3390/ijms21165652

**Published:** 2020-08-06

**Authors:** Mette G. Christensen, Nanna Johnsen, Marianne Skals, Aimi D. M. Hamilton, Peter Rubak, Anne-Mette Hvas, Helle Praetorius

**Affiliations:** 1Department of Biomedicine, Aarhus University, 8000 Aarhus C, Denmark; mgch@biomed.au.dk (M.G.C.); nanna-johnsen@biomed.au.dk (N.J.); msk@nmdpharma.com (M.S.); aiha@biomed.au.dk (A.D.M.H.); 2Clinic for Diagnostics, Aalborg University Hospital, 9000 Aalborg, Denmark; peterrubak@gmail.com; 3Department of Clinical Biochemistry, Aarhus University Hospital, 8000 Aarhus C, Denmark; annehvas@rm.dk

**Keywords:** P2Y_1_, P2Y_12_, sepsis, *Escherichia coli*, HlyA, thrombocytes

## Abstract

Urosepsis is a potentially life-threatening, systemic reaction to uropathogenic bacteria entering the bloodstream of the host. One of the hallmarks of sepsis is early thrombocyte activation with a following fall in circulating thrombocytes as a result of intravascular aggregation and sequestering of thrombocytes in the major organs. Development of a thrombocytopenic state is associated with a poorer outcome of sepsis. Uropathogenic *Escherichia coli* frequently produce the pore-forming, virulence factor α-haemolysin (HlyA), of which the biological effects are mediated by ATP release and subsequent activation of P2 receptors. Thus, we speculated that inhibition of thrombocyte P2Y_1_ and P2Y_12_ receptors might ameliorate the septic response to HlyA-producing *E. coli*. The study combined in vitro measurements of toxin-induced thrombocyte activation assessed as increased membrane abundance of P-selectin, fibronectin and CD63 and data from in vivo murine model of sepsis-induced by HlyA-producing *E. coli* under infusion of P2Y_1_ and P2Y_12_ antagonists. Our data show that the P2Y_1_ receptor antagonist almost abolishes thrombocyte activation by pore-forming bacterial toxins. Inhibition of P2Y_1_, by constant infusion of MRS2500, markedly increased the survival in mice with induced sepsis. Moreover, MRS2500 partially prevented the sepsis-induced depletion of circulating thrombocytes and dampened the sepsis-associated increase in proinflammatory cytokines. In contrast, P2Y_12_ receptor inhibition had only a marginal effect in vivo and in vitro. Taken together, inhibition of the P2Y_1_ receptor gives a subtle dampening of the thrombocyte activation and the cytokine response to bacteraemia, which may explain the improved survival observed by P2Y_1_ receptor antagonists.

## 1. Introduction

Sepsis is a life-threatening condition with an overwhelming immune response to infection. Unfortunately, the number of patient admissions under the diagnosis sepsis is markedly increasing [1], frequently with the urinary tract as the primary infection site [2,3]. Urinary tract infections are regularly caused by *Escherichia coli* (for review, see [4]) in both the primary and secondary health sector. The *E. coli* that causes severe infections, such as pyelonephritis, are known to vary serotypically from *E. coli* in the intestinal microbiome [5,6,7] and produce a variety of virulence factors [5,6,7,8,9]. Of those, α-haemolysin (HlyA) is the most frequent in clinical isolates [10] and is thus, associated with the ability to create and sustain severe infection. Our previous data support that HlyA is largely responsible for the septic symptoms observed in response to bacteraemia with uropathogenic *E. coli* [11]. Interestingly, the biological effects of HlyA are intimately associated with extracellular adenosine triphosphate (ATP) signalling [12,13,14,15,16]. This is a consequence of ATP being released in a non-lytic fashion directly through the pore created by HlyA [17] and subsequent activation of P2 receptors [12,13,14,15,16,18]. Since ATP is a renowned damage-associated molecular pattern (DAMP) molecule [19], it is thus an aggressive activator of immune cells and a prime candidate for promoting the cytokine rush during septic shock. In this context, it is interesting that the phenomenon of ATP-mediated amplification of virulence is not unique for HlyA, but is a feature for several other bacterial pore-forming toxins [20,21,22,23,24] and membrane attack complex from the complement system [25]. This essentially implies that P2 receptor signalling is likely to be a more general issue in sepsis.

Sepsis and septic shock are associated with a very early reduction in circulating thrombocytes, and thrombocytopenia is generally associated with a poorer outcome in sepsis [26,27]. This does not immediately imply that a low level of circulating thrombocytes, per se, is responsible for the poorer outcome of sepsis. However, the sepsis-induced reduction in circulating thrombocytes reflects both organ accumulation but is also a consequence of microthrombus formation [28]. This obstruction of the microcirculation is responsible for the reduced tissue oxygenation and multiorgan failure seen during severe sepsis (for overview, see [29]). The drop in circulating thrombocytes is easily confirmed in our sepsis model, a process markedly accelerated by the presence of HlyA [11]. Thus, it is tempting to speculate that HlyA-induced ATP release and subsequent intravascular degradation to adenosine diphosphate (ADP) would contribute to escalate the thrombocyte activation and thereby increase the mortality in *E. coli*-induced sepsis.

Thrombocytes express three P2 receptor subtypes: P2X_1_, P2Y_1_ and P2Y_12_ (for review, see [30]). ATP is an agonist for both the P2X_1_ and P2Y_1_ receptors [31,32], whereas P2Y_12_ and P2Y_1_ receptors are responsible for the renowned ADP-mediated thrombocyte activation [33,34,35]. Thus, it would be logical to target either of the thrombocyte P2 receptors to prevent sepsis-induced thrombocyte activation. Interestingly, mice lacking the P2X_1_ receptor are protected against the sepsis-induced cytokine rush [36]. However, specific inhibition of P2X_1_ receptors causes an acute reduction of circulating thrombocytes in itself and, thus, subsequently decreases the overall survival of urosepsis [37]. Therefore, the P2X_1_ receptor is not immediately a promising target to reduce thrombocyte activation during sepsis. Here, we hypothesise that interference with either P2Y_1_ or P2Y_12_ may dampen the thrombocyte activation and improve the survival in a murine model of urosepsis.

The data demonstrate that thrombocyte activation in vitro with pore-forming toxins requires P2 receptor activation and is essentially abolished by inhibition of P2Y_1_, whereas inhibition of P2Y_12_ only moderately reduced the thrombocyte activation. Moreover, constant infusion of a P2Y_1_ receptor antagonist in a murine model of urosepsis markedly reduced the overall mortality. This finding is associated with a less pronounced reduction in circulating thrombocytes in response to sepsis and a reduced proinflammatory response to a given load of circulating *E. coli*. Interestingly, infusion of a P2Y_12_ receptor antagonist, cangrelor, showed a tendency towards improving survival, even though this was not statistically significant. Similar to inhibiting the P2X_1_ receptor [37], cangrelor reduced the plasma levels of proinflammatory cytokines. Thus, we can conclude that inhibition of P2Y_1_ receptors improves the survival of sepsis induced by HlyA-producing *E. coli* possibly by dampening the thrombocyte and cytokine response.

## 2. Results

Pore-forming bacterial toxins are intimately associated with release of intracellular ATP to the extracellular compartment. The biological action of ATP is associated with P2X or P2Y receptor activation on various cell types, such as erythrocytes, monocytes and renal epithelial cells [12,13,16,20,22]. The pore-forming bacterial toxin, α-toxin from *S. aureus* has previously been shown to be able to initiate thrombocyte aggregation [38]. Therefore, we tested whether thrombocyte P2Y_1_ and P2Y_12_ receptors are involved in the thrombocyte activation triggered by bacterial pore-formers. To test this, thrombocyte activation was measured by flow cytometry as a combined increase in fibronectin, CD63 and P-selectin positive thrombocytes as previously described [39]. Figure 1A shows the level of P-selectin expression on thrombocytes in healthy volunteers in relation to the maximal thrombocyte activation achieved with 140 µM ADP or 371 µM thrombocyte activating peptide (TRAP) as positive controls. With this technique, we could easily verify that α-toxin from *Staphylococcus aureus* markedly activated human thrombocytes both in whole blood (Figure 1A) and in platelet enriched plasma (data not shown). Our data demonstrate that thrombocyte activation was not exclusive to α-toxin but could be recapitulated with leukotoxin A (LtxA) from *Aggregatibacter actinomycetemcomitans*, which similarly to HlyA belongs to the repeat-in-toxin (RTX) family of bacterial pore-formers (Figure 1A). In high concentrations, both toxins activated the thrombocytes to a level similar to the positive controls (data not shown). However, the concentration of pore-forming toxins was adjusted to gain approximately 25% thrombocyte activation to allow detection of both inhibitory and potentiating effects (Figure 1A). The in vitro studies could unfortunately not be performed by HlyA itself. HlyA requires Ca^2+^ to insert into biological membranes, and, because addition of Ca^2+^ alone caused a sizable and very variable thrombocyte activation, it was not possible to quantitate the HlyA response in a reliable manner. However, our previous studies have demonstrated that, in terms of ATP-dependent amplification of the biological effect of HlyA, α-toxin and LtxA are quite similar [12,17,18,20,22]. Figure 1B verifies that ADP-dependent thrombocyte activation indeed is inhibited by selective antagonists towards the two thrombocyte P2Y receptors, P2Y_1_ (MRS2500) and P2Y_12_ (PSB0739), used in the study.

Consistent with our previous studies on erythrocytes, the biological effects of α-toxin and LtxA was markedly inhibited by P2 receptor inhibition on thrombocytes [20,22]. Figure 1C shows that MRS2500 (9 μM) reduced the thrombocyte activation induced by LtxA considerably (around 80%) regardless of whether the readout was fibronectin, CD63 or P-selectin. PSB0739 (18 μM) showed the same tendency, with ~60% inhibition of the LtxA-induced thrombocyte activation, but the results were not statistically significantly different from samples exposed to LtxA alone (Figure 1C). Combining the antagonists only had a marginally additional effect to MRS2500 alone. Similar results were found with α-toxin, only here the effect of PSB0739 was even less pronounced (~55% inhibition, Figure 1D). Thus, we can confirm that the biological effect on thrombocytes of two pore-forming toxins largely is caused by secondary purinergic signalling. Since the data were obtained from whole blood samples, it is reasonable to assume that the ADP activating the thrombocytes originated from ATP released from erythrocytes in response to the pore-forming toxins [20,22]. Therefore, we tested whether the toxins could activate the thrombocytes directly and whether this effect was P2Y receptor-dependent. In platelet enriched plasma (PRP), the pore-forming toxins α-toxin and LtxA still activated the thrombocytes. Similar to what was shown in whole blood, the P2Y receptor antagonists still lowered the thrombocyte activation in response to the pore-forming toxins (20–40% reduction, Figure 1E). However, the overall effect was less pronounced compared to whole blood samples. This potentially suggests that the effect of pore formation was less amplified by ATP release and subsequent P2 receptor activation when thrombocytes are not surrounded by other blood cells. This supports the notion that thrombocyte activation by the pore-forming toxins in whole blood is likely to be amplified by ATP released from erythrocytes or other of the formed blood components.

The question is whether this finding has any bearing in vivo. To test this, we used a murine sepsis model with direct injection into the tail vein of *E. coli* producing the pore-forming toxin α-haemolysin (HlyA) [11,36]. Our previous studies have demonstrated that HlyA secreted from uropathogenic *E. coli* constitute a marked virulence factor solely responsible for the septic response during bacteraemia [11]. We demonstrated that the mice only developed septic symptoms when the HlyA operon was expressed in an *E. coli* strain (K12) without any other important virulence factors. Figure 2A shows that constant infusion with the P2Y_1_ receptor antagonist MRS2500 (25.9 μg/h) compared to saline infusion increased the survival of mice exposed to the exact same HlyA-producing *E. coli* strain (*p* = 0.048). We confirm that the induction of sepsis is associated with a substantial drop in circulating thrombocytes (Figure 2B). In this controlled sepsis model, the reduction in circulating thrombocytes is likely to reflect the degree of intravascular thrombocyte activation [40,41]. Interestingly, the mice infused with MRS2500 did not exhibit the reduction in thrombocytes typically observed in response to sepsis with uropathogenic *E. coli* in this mouse model (Figure 2B) [11,36]. In mice exposed to saline infusion, the number of thrombocytes drops to 56% after 2.5-h exposure to 30 × 10^6^
*E. coli* (*p* = 0.005). However, in mice infused with MRS2500, this reduction was no longer statistically significant (22% reduction after 2.5 h, *p* = 0.600, Figure 2B). Of note, the mice infused with MRS2500 developed haemoglobinuria earlier (after 168 min) than mice exposed to saline infusion (after 253 min, *p* = 0.070, Figure 2C) despite having the same degree of intravascular haemolysis (*p* = 0.340, Figure 2D) and activation of intravascular coagulation (Figure 2E). Thus, P2Y_1_ receptor inhibition seemingly ameliorates some of the septic symptoms in mice directly without affecting the severity of the infection.

We speculated that lower mortality in mice exposed to MRS2500 might result from a dampening of the proinflammatory cytokine response since high levels of proinflammatory cytokines are associated with a negative outcome of sepsis [42]. However, to our surprise, MRS2500 only had a minor effect on the cytokine response to infection with HlyA-producing *E. coli* per se. Figure 3 demonstrates the well-documented increase in plasma levels of the proinflammatory cytokines tumour necrosis factor factor α (TNF-α), interleukin-1β (IL-1β), IL-6 and the mouse ligand for the IL-8 receptor, keratinocyte chemoattractant (KC) in response to iv-injection of the HlyA-producing *E. coli* [11,36]. The response was seemingly similar in mice exposed to MRS2500 and saline infusion, except for KC where the plasma-levels were statistically significantly higher in mice infused with MRS2500 (Figure 3B, *p* = 0.0005). However, when the cytokine response was evaluated relative to the measured intravascular bacterial load, the proinflammatory cytokines tended to be lower at a given stage of infection in mice exposed to MRS2500 compared to mice exposed to saline (Figure 4B–E). This was the case for all the proinflammatory cytokines (IL-6, *p* = 0.0002; IL-1β, *p* = 0.005; TNF-α, *p* = 0.033) except KC (*p* = 0.472).

Notably, the improved survival of mice infused with MRS2500 did not result from a more effective clearance of bacteria from the blood. MRS2500 infusion actually resulted in a statistically significantly higher number of circulating bacteria in the blood after 2.5 h compared to mice exposed to saline infusion (Figure 4A, *p* = 0.03). Thus, constant infusion of the P2Y_1_ receptor antagonist increases the survival of mice with acute urosepsis despite a slightly higher level of circulating bacteria.

In a clinical context, the P2Y_12_ receptor is a more established target for pharmacologic inhibition of thrombocyte activation. However, our in vitro data show a larger effect of P2Y_1_ receptor inhibition on thrombocyte activation compared to a P2Y_12_ receptor antagonist after exposure to pore-forming bacterial toxins. To investigate whether this pattern was also reflected in the in vivo data, we infused the P2Y_12_ receptor antagonist, cangrelor, in mice exposed to HlyA-producing *E. coli*. Figure 5A shows that cangrelor at an infusion rate of 8.6 μg/h (low dose) had a tendency towards improved survival, although it did not reach statistical significance (*p* = 0.070). Therefore, we increased the infusion rate to 86.6 μg/h, which counteracted the potential slight beneficial effect seen at the lower dose (Figure 5A). Similarly, cangrelor at 8.6 μg/h was not able to prevent the bacteria-induced reduction in circulating thrombocytes (~50% reduction in both groups, Figure 5B), and there was no observable difference in either the intravascular haemolysis (*p* = 0.800, Figure 5D) or the onset of haemoglobinuria (142 min (saline) and 93 min (cangrelor), *p* = 0.700, Figure 5C). Neither did we detect any difference in the bacteria-induced activation of intravascular coagulation measured as formed thrombin–antithrombin (TAT) complexes in the absence or presence of cangrelor (8.6 μg/h, Figure 5E). However, the trend that inhibiting thrombocyte P2 receptors diminished the proinflammatory cytokine response was also observed when infusing cangrelor. Figure 6 shows that the plasma levels of KC were significantly lower (Figure 6B, *p* = 0.017) in septic mice infused with cangrelor (8.6 μg/h) compared to those infused with saline. The same pattern was seemingly true for IL-1β (Figure 6C, *p* = 0.065) and IL-6 (Figure 6D, *p* = 0.066), where the cangrelor infused plasma levels were on the rim of being statistically significantly lower than the saline controls. In contrast to P2Y_1_-antagonist infusion, we did not find the number of circulating bacteria after 2.5 h to be different with or without cangrelor (*p* = 0.2, Figure 7A). When the proinflammatory cytokines were expressed relative to the number of circulating bacteria, the cytokine levels were no longer statistically significantly different between the two groups (Figure 7B–E). These data suggest that inhibition of P2Y_12_ receptors, at best, gives a marginal effect on the outcome of sepsis.

These data could suggest that there might be an additional beneficial effect by combining P2Y_1_ and P2Y_12_ receptor inhibition during the induced sepsis. Thus, we tested whether combined infusion of MRS2500 (25.9 μg/h) and cangrelor (8.6 μg/h) would give a more advantageous outcome. Unfortunately, this combination proved to be quite the opposite. The animals infused with the combination died very quickly (within an hour) with extreme bleeding tendency and a marked potentiation of the septic symptoms (data not shown).

## 3. Discussion

Sepsis constitutes the host reaction to a circulating infectious agent that potentially leads to life-threatening overstimulation of the immune system and a concomitant hypercoagulable state with a massive micro-thrombus formation and subsequent reduction of the microcirculation. Urinary tract infections are frequently the primary infection-site in patients with sepsis [3], and, thus, *E. coli* are found in around 30–40% of blood cultures from patients with suspected sepsis [43]. The *E. coli* that cause severe infection regularly produce a variety of virulence factors, of which HlyA is constantly found in clinical isolates [10]. As a virulence factor, HlyA is known to increase the risk of developing sepsis in response to a peripheral infection [44,45,46,47]. We were, however, surprised to find that this effect not only increased the invasion of bacteria to the bloodstream from peripheral infectious sites but also markedly accelerated all the septic symptoms once the bacteria was present in the blood [11]. We tested this by means of a non-HlyA-expressing *E. coli* strain K12, transfected with a plasmid coding for the entire *hlya* operon or with the same plasmid with loss of function deletion (201–2173 bp) in *hlya* [11]. Mice exposed to the HlyA-producing strain died early with massively increased proinflammatory cytokines, intravascular haemolysis, haemoglobinuria, thrombocytopenia and intravascular coagulation, whereas the mice exposed to an equal number of non-HlyA-producing K12 survived the full observation period with little or no changes in the sepsis parameters [11]. Since the biological effect of HlyA to a high extend is secondary to ATP release and P2 receptor activation [12,14,16,17], we were interested in whether interfering with thrombocyte activation would improve the outcome of sepsis with HlyA-producing *E. coli*.

The included in vitro data show that thrombocyte activation by pore-forming bacterial toxins requires P2Y receptor activation, with P2Y_1_ as the predominant receptor mediating the activation. Pore-forming bacterial toxins have previously been shown to activate thrombocytes [48], and, with our previous data regarding the concomitant non-lytic ATP release, it is not exceedingly surprising. Nevertheless, these data, first of all, underscore that the biological effect of pore-forming toxins takes the flavour of the P2 receptor expression of the given target cell or tissue. Moreover, the data suggest that the erythrocytes are the main source of ATP in response to bacterial toxins because the thrombocyte activation by the toxins was much more efficient in whole blood compared to platelet enriched plasma. HlyA can, in principle, insert into any biological membrane, and, thus, any blood cell could be the source of ATP. However, the insertion of HlyA is a stochastic event, and therefore, by chance, HlyA is more likely to encounter an erythrocyte. Interestingly, the data also suggest that P2Y_1_ receptor inhibition was more efficient in preventing thrombocyte activation. One explanation could be the difference in the preferred agonist profile of the two receptors. Where ADP is the prime agonist for the P2Y_12_ receptor, both ATP and ADP work as agonists for the P2Y_1_ receptor [31,32]. Since an ADP-dependent P2 receptor activation in the case of bacterial toxins would be secondary to degradation of ATP, it is likely that a receptor directly activated by the released ATP would take precedence in the following response.

Interestingly, in our sepsis model, continuous infusion of a P2Y_1_ receptor antagonist markedly increased the survival of the mice. Our model is proven to be exceedingly reliable for monitoring the development of septic symptoms in response to bacteraemia. Urosepsis is not easily modelled since mice do not readily develop sepsis after installation of bacteria in the urinary bladder. Therefore, our group has established a model that mimics a fast-developing urosepsis, as it may be seen after instrumentation of the urinary tract [11,36], to examine the effect of P2 receptors upon HlyA-induced ATP release for the septic response. P2 receptor activation has a substantial impact on the course of sepsis, and generally, for all P2 receptors tested so far, lack of or inhibition accelerates the septic symptoms and increase the mortality [36,37]. Thus, P2Y_1_ receptor inhibition is the first to show an increased survival of *E. coli*-induced sepsis, and, therefore, the effect of MRS2500 on survival is likely to result exclusively from P2Y_1_ receptor inhibition. However, we cannot know which cell type is explicitly responsible for the positive effect on survival. We know that MRS2500 does not affect the HlyA induced haemolysis [12], and, thus, the effect of MRS2500 is not an indirect effect on the erythrocytes. However, P2Y_1_ receptors are expressed on various cells in the vascular compartment [49], and, thus, the increased survival could, in principle, result from effects on the endothelial cells or other immune cells. The expression profile for P2Y_12_ is (for review, see [50]), however, much more restrictive and is primarily confined to the thrombocytes. Therefore, it is interesting to observe that cangrelor tendentially improve the survival of sepsis even though this is not statistically significant. This indirectly supports that the effect of MRS2500 on survival may be mediated through interference with thrombocyte activation. Notably, at higher doses of cangrelor, this marginal effect is lost, which likely reflects off-target effects on other P2 receptors. In terms of the disappointing effect of cangrelor on the survival, our data here support previous studies on intravascular coagulation activated by lipopolysaccharide (LPS) in humans [51]. This study is a randomised, double-blinded, placebo-controlled study, which could not find any effect of P2Y_12_ receptor inhibition on LPS-induced intravascular coagulation.

The substantiated protocol to precisely detect human thrombocyte activation is not immediately adaptable to mice. However, if one allows the notion that the acute *E. coli*-induced reduction in circulating thrombocytes may partially reflect intravascular thrombocyte activation, the in vivo data do support that P2Y_1_ receptor antagonist has an effect on the thrombocytes during bacteraemia. Thus, our data confirm that intravenous injection of *E. coli* produces an early and substantial decrease in the thrombocyte count [11,36]. However, in the presence of the P2Y_1_ receptor antagonist, this response is much more variable and no longer statistically significant. This is consistent with the P2Y_1_ receptor not being the only thrombocyte receptor activated by HlyA and subsequent ATP release. However, the data do underscore thrombocytes as one of the critical targets for a P2Y_1_ receptor antagonist in the bloodstream.

Notably, the reduced thrombocyte response to P2Y_1_ is not mirrored by a reduction in intravascular coagulation, which could potentially indicate that the decrease in thrombocyte numbers may not only be an effect of intravascular coagulation and microthrombus formation. Notably, injection of LPS is in itself sufficient to reduce the number of circulating thrombocytes, similar to what is observed during sepsis [52]. Interestingly, this LPS-induced reduction in circulating thrombocytes is also prevented by a P2Y_1_ receptor antagonist, whereas this is not the case for either P2X_1_ or P2Y_12_ receptor antagonists [52]. The authors find that the drop in circulating thrombocytes after LPS-exposure primarily results from thrombocyte translocation to the periphery tissues like the lung [52]. However, it must be noted that, in our model of urosepsis, the reduction in thrombocytes is not induced by LPS because the K12 strain with a deletion in *hlya* did not affect the number of circulating thrombocytes [11]. As mentioned, HlyA is known to trigger non-lytic ATP release from cells directly through the formed pore [17] and thus provides a direct link to the P2Y_1_ receptor activation. In the case of LPS, the link to P2Y receptor activation is not that straightforward but presumably involves Toll-Like Receptor 4 (TLR-4)-activation [53] and subsequent thrombolytic release of dense granules and thus ADP. Regardless, there is substantial evidence for P2Y_1_ receptor activation being central for the reduction in circulating thrombocytes as observed during sepsis. It has previously been speculated that combined inhibition of P2Y_12_ and P2Y_1_ receptor may show a benefit in some clinical situations compared to inhibition of either of the receptors alone [54]. We did, however, test the combined effect of inhibition of both receptor subtypes and found that the combination had a detrimental effect and accelerated the time of death upon intravenous exposure to uropathogenic *E. coli*.

Remarkably, inhibition of thrombocyte P2 receptors during sepsis seems to slightly dampen the cytokine response induced by the bacteria and their virulence factors. In the case of P2Y_1_, this was first obvious when the level of proinflammatory cytokines was evaluated relative to the number of circulating bacteria. This is a relevant measure because HlyA has been shown to markedly accelerate the cytokine production in this sepsis model [11] and because HlyA is only produced by live bacteria. In the case of P2Y_12_, the pattern was similar, with a less aggressive rise in the proinflammatory cytokines during sepsis. This is potentially important because mice lacking the P2X1 receptor, also show a very dampened cytokine response compared to control [37]. These findings support the notion that thrombocytes participate in the early immunological response to agents in the circulation [55]. This has been suggested to result from thrombocytes participating in the recruitment of neutrophils [56] and in the activation of neutrophils to release neutrophil extracellular traps (NETs) [57]. In Gram-negative sepsis with *E. coli*, NETs are induced by activation of TLR-4 on thrombocytes, that subsequently activates neutrophils via CD11a [58]. Generation of NETs is beneficial during sepsis because it can capture bacteria in the bloodstream, and thereby reduce the number of circulating bacteria [59]. This may potentially explain why we can detect a marginally, but statistically significant, increase in the number of circulating bacteria during infusion with MRS2500.

However, the dampening of the cytokine response may very well be closely associated with the increased survival observed after infusion with P2Y_1_ receptor antagonist. The exceedingly high plasma levels of proinflammatory cytokines have previously been termed *the cytokine storm* and are known as a hallmark of severe sepsis and septic shock [42]. This auto-amplification of the immunological response is associated with a poorer outcome, but, unfortunately, attempts to dampen the response have failed in clinical trials [42]. Our sepsis model nicely reproduces elements of *the cytokine storm,* with exceedingly high plasma levels of proinflammatory cytokines, and, in our model, we too observed that high cytokine-levels are associated with poorer outcome [36]. If indeed the proinflammatory cytokine propels the pathogenesis during septic shock, then P2Y_1_ receptor inhibition might dampen the escalation. However, if thrombocytes were an easy early target to dampen the immune response and improve the survival of sepsis, one would potentially have expected larger responses. The question is if interference with thrombocyte function will prove to be an effective target in the treatment of sepsis, particularly because of the detrimental bleeding complications observed when targeting P2Y_1_ and P2Y_12_ in combination. Our safest bet with regard to supportive therapy would be to work on a strong, selective P2Y_1_ receptor antagonist. This may well prove to safeguard and keep the thrombocytes in circulation and simultaneously restrain cytokine auto-amplification induced by the bacteraemia.

## 4. Materials and Methods

### 4.1. Animals

Experiments were performed on male Balb/cJ mice from Janvier Labs (Saint-Berthevin, France). All animals were kept at the Department of Biomedicine, Aarhus University, Denmark, and experiments were approved by the Danish ethic committee for animal research “Dyreforsøgstilsynet” (2014-15-0201-00316).

### 4.2. Bacteria

The *E. coli* strain used for the experiments was kindly provided by Professor Rodney Welch (University of Wisconsin, Madison, WI, USA). The *E. coli* strain WAM1824 was constructed in the K12 background strain (WAM1808) [60]; a strain that does not produce HlyA and without any other virulence factors. For the HlyA-producing WAM1824 variant, the WAM1808 strain was transfected with the plasmid pWAM582, which contains the entire HlyA operon and a chloramphenicol resistance gene for selection.

WAM1824 was grown on blood agar plates supplemented with chloramphenicol (20 µg mL^−1^) and kept at 4°C for up to one month. Before each experiment, an overnight culture was produced by transferring one colony to 4 mL Lysogeny-Broth (LB)-medium containing chloramphenicol (20 µg mL^−1^) for selection. This preparation incubated overnight at 37°C and 250 rpm. The overnight culture (3 mL) was centrifuged at 1162× *g* for 10 min, followed by a wash in saline (0.9%), and centrifuged 5 min at 1162× *g*. The bacterial pellet was re-suspended in sterile saline (0.9%), and bacteria were counted on flow cytometry (BD Accuri 6, BD Biosciences, Franklin Lake, NJ, USA).

### 4.3. Induction of Sepsis in Mice

Male Balb/cJ mice (8–10 weeks, 24.6 ± 0.2 g) were anaesthetised by subcutaneous injection of ketamine (100 mg/kg) and xylazine (10 mg/kg) and placed on a 38°C heating plate. The mice were kept under anaesthesia for the duration of the experiment. The amount of anaesthesia was adjusted to the individual need, monitored by response to stimulation of whiskers and extremities. The mice were injected through the tail vein with 150 μL of either 30 × 10^6^
*E. coli* in solution with P2Y receptor antagonist and chloramphenicol (33 μg) or 30 × 10^6^
*E. coli* in saline solution with chloramphenicol (33 μg). Injections were carried out with a needle (27G) attached to a syringe via fine tubing. The mice received a bolus injection with either the P2Y_1_ antagonist MRS2500 (0.366 μg/g) or the P2Y_12_ antagonist cangrelor (either 0.06 μg/g or 0.6 μg/g) and after that continuous infusion with MRS2500 (25.9 μg/h), cangrelor (8.6 μg/h or 86.4 μg/h) or sterile saline (66 μL/h) throughout the experiment. For survival data, the mice were observed for up to 6 h and the onset of haemoglobinuria and time of death noted. For thrombocyte count, cytokines, TAT-complexes, haemolysis and blood culture, the mice were terminated after 2.5 h, and a blood sample was drawn from the inferior vena cava with a citrate-containing syringe.

### 4.4. Bacterial Load in Septic Mice

A fixed volume of 5 μL whole blood was diluted in 45 μL sterile saline and streaked on an agar plate. The agar plate was incubated overnight, and the number of colony-forming units (CFU) was counted.

### 4.5. Thrombocyte Count in Septic Mice

Whole blood (5 μL) was incubated for 15 min with 2 μL CD42d antibody (BD Biosciences, Franklin Lake, NJ, USA) and 60 μL PBS. Secondary fluorescein isothiocyanate (FITC)-conjugated antibody (2 μL, BD Bioscience) was added and incubated another 15 min in the dark; 20 μL sample was transferred to 1.5 mL formaldehyde (0.02%), and CD42-d-FITC positive cells (thrombocytes) were counted on flow cytometer (BD Accuri 6, Franklin Lake, NJ, USA).

### 4.6. Measurement of Haemolysis

Blood samples were centrifuged immediately after collection for 10 min at 1162× *g*, and plasma was diluted 1:8 with saline before measurement of the absorbance at 410 nm on a spectrophotometer (Ultraspec III, LKB Biochom). The remaining plasma was stored at −20 °C for later analysis of cytokines and TAT complexes.

### 4.7. Thrombin–Antithrombin (TAT) Complex

TAT was measured in plasma samples using TAT Complexes Mouse Elisa Kit from Abcam (Cambridge, UK). Measurements were done according to manufactures instructions. Plasma was stored for up to 6 months at −20 °C prior to TAT measurements.

### 4.8. Measurement of IL-6, IL-1β, TNF-α and KC

Plasma for cytokine measurements was isolated from a blood sample centrifuged for 10 min at 1162× *g*. Plasma samples were kept at −20 °C for up to 6 months for analysis. Levels of IL-6, IL-1β, TNF-α and keratinocyte chemoattractant (KC, murine equivalent to human IL-8) were measured by flow cytometry (BD Accuri 6, Franklin Lake, NJ, USA) with a cytometric bead array flex set from BD Biosciences according to the manufacturer’s instruction.

### 4.9. Measurement of In Vitro Activation of Human Thrombocytes

Blood samples were obtained from healthy volunteers with a 21-gauge needle into a 4 mL citrated tubes (3.2%) after discarding the first 4 mL whole blood extracted. All human donors gave their written content, and the study was approved by the Danish Scientific Ethics Committee (M201100217). Platelet-rich plasma was obtained by centrifugation of whole blood at 200× *g* for 15 min at room temperature. Samples for both whole blood and PRP analysis were left untouched for one hour to equilibrate. Activation of thrombocytes was determined by a method previously described by Rubak et al. [39]. In short, 5 µL of each antibody specific for the various thrombocyte activation markers CD42b-PE (AH diagnostics, Tilst, Denmark), CD63-PCy7 (BD Bioscience, San Jose, CA, USA), CD62P-APC (P-selectin, eBioscience, San Diego, CA, USA) and anti-fibrinogen-FITC (Diapensia HB, Linköping, Sweden) were pooled in tubes with a HEPES buffered salt solution (HBS, 30 µL) along with 5 µL activating solution and/or 5 µL P2Y-receptor antagonist. Five microlitres whole blood or PRP were added to the antibody pool, and the mixture was incubated for 10 min at room temperature in the dark. The samples were fixed with 0.02% formaldehyde before analysis on Navios flow cytometry (Beckman Coulter, Miami, Florida). As thrombocyte activator we used ADP (140 µM, Sigma-Aldrich, Søborg Denmark), thrombocyte activating peptide (TRAP, 371 µM, JPT Peptide Technologies, Berlin, Germany), α-toxin from *S. aureus* (Sigma-Aldrich, Søborg Denmark) and leukotoxin (LtxA) from *A. actinomycetemcomitans* (purified as previously described [61]), whereas the thrombocyte P2 receptors were inhibited by the P2Y_1_ receptor antagonist MRS2500 (Tocris, Bristol, UK) and/or the P2Y_12_ receptor antagonist PSB0739 (Tocris, Bristol, UK). The concentration of α-toxin and LtxA added was adjusted to cause approximately 25% activation of the thrombocytes. Negative controls were conducted by exchanging the agonist and receptor antagonist with an EDTA-HEPES buffer (6 mM, Sigma-Aldrich, Søborg, Denmark), whereas a sample with pure HEPES buffer was used as a measure of thrombocyte pre-activation.

### 4.10. Statistics

Statistical analyses were performed using GraphPad Prism. The data were tested for normal distribution by the Kolmogorov–Smirnov test. The in vitro data on thrombocyte activation were analysed by one-way ANOVA (Kruskal–Wallis with Dunn’s post-test). Survival and studies of haemoglobinuria were analysed by Kaplan–Meier plots. Normally distributed data from the in vivo experiments were analysed with unpaired *t*-test between sepsis groups treated with either saline or MRS2500/Cangrelor. Data that were not normally distributed were analysed with Mann–Whitney test. Data from thrombocyte counts and TAT complexes were analysed with two-way ANOVA. All data are presented as mean ± standard error of the mean, and the data were considered statistically significantly different when the *p*-value was less than 0.05.

## Figures and Tables

**Figure 1 ijms-21-05652-f001:**
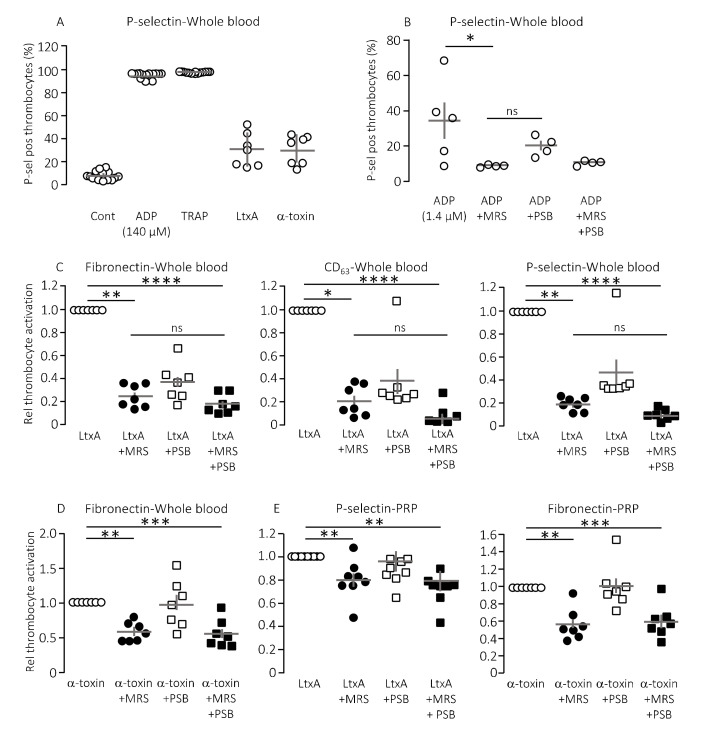
The effect of P2Y receptor blockade on the thrombocyte activation induced by pore-forming toxins in vitro. Thrombocyte activation is measured as the P-selectin, fibronectin and CD63 positive cells as a percentage of the total number of thrombocytes. (**A**) The pore-forming toxin α-toxin (*n* = 7) and leukotoxin A (LtxA, 30 μg mL^−1^, *n* = 7) readily activates thrombocytes here shown as the relative number of P-selectin positive thrombocytes. Activation of thrombocytes with adenosine diphosphate (ADP, 140 µM, *n* = 14) and thrombocyte activating peptide (TRAP, 371 µM, *n* = 14) was used as positive controls and a HEPES-buffered solution was used as negative control (Cont). The addition of pore-forming toxin was adjusted to give a thrombocyte activation of approximately 25%. (**B**) The effect of P2Y_1_ and P2Y_12_ receptor inhibition on the ADP-induced thrombocyte activation. MRS2500 (MRS) was used in a concentration of 9 µM and PSB0739 (PSB) 18 µM. (**C**) The effect of MRS2500 9 µM (*n* = 7) and PSB0739 18 µM (*n* = 7) on the thrombocyte activation induced by LtxA in whole blood measured as fibronectin, CD63 and P-selectin positive thrombocytes. For the combined effect of MRS2500 and PSB0739 *n* = 7. (**D**) The effect of MRS2500 9 µM (*n* = 7) and PSB0739 18 µM (*n* = 7) on α-toxin-induced thrombocyte activation in whole blood. For the combined effect of MRS2500 and PSB0739(*n* = 7). (**E**) The effect of MRS2500 9 µM (*n* = 8) and PSB0739 18 µM (*n* = 8) on LtxA-induced or α-toxin-induced thrombocyte activation in platelet enriched plasma (PRP). For the combined effect of MRS2500 and PSB0739 *n* = 8. All data are given as single observations and mean ± S.E.M.* *p* < 0.05, ** *p* < 0.01, *** *p* < 0.001 and **** *p* < 0.0001.

**Figure 2 ijms-21-05652-f002:**
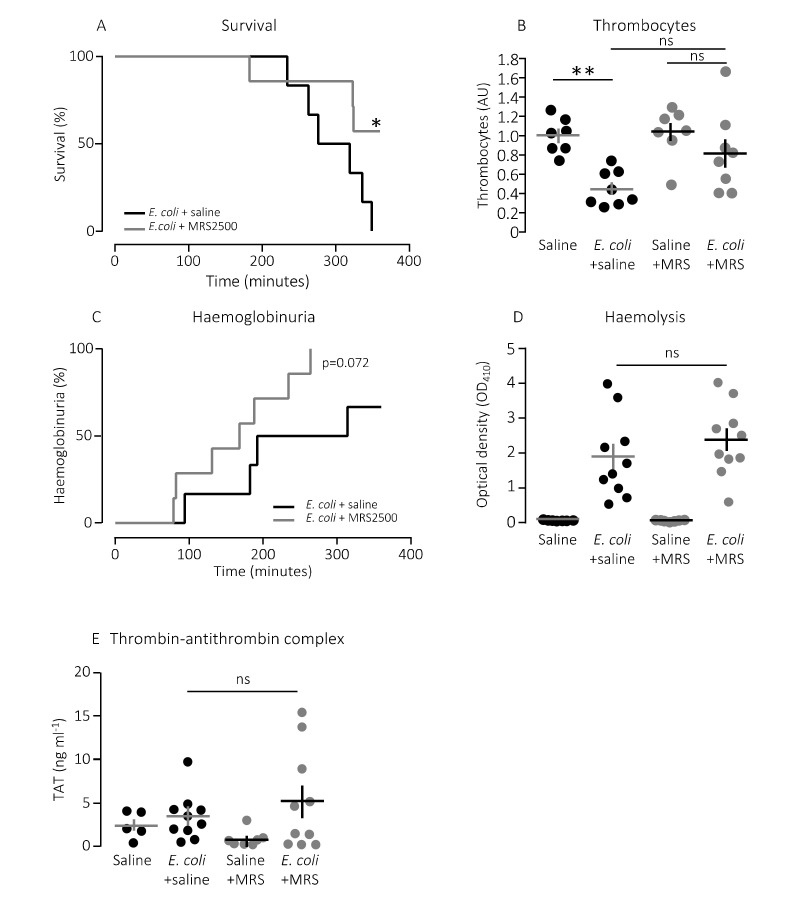
P2Y_1_ receptor antagonist MRS2500 (MRS) increase survival of mice exposed to sepsis with *E. coli*. (**A**) Mice with bacteraemia receiving continuous infusion of either saline (*n* = 6) or the P2Y_1_ receptor antagonist MRS2500 (25.9 μg/h, *n* = 7) for up to 6 h. The mice subjected to bacteraemia received 30 × 10^6^
*E. coli* (WAM1824) iv. Data are shown as Kaplan–Meyer plot, * indicates *p* < 0.05. (**B**) The relative number of thrombocytes 2.5 h after injection of 30 × 10^6^
*E. coli* iv (saline *n* = 7, *E. coli* + saline *n* = 8, saline + MRS2500 (25.9 μg/h) *n* = 7, *E. coli* + MRS2500 (25.9 μg/h) *n* = 8). (**C**) Development of haemoglobinuria in mice exposed to 30 × 10^6^
*E. coli* for up to 6 h infused with either saline (*n* = 6) or the P2Y_1_ receptor antagonist MRS2500 (25.9 μg/h, *n* = 7). (**D**) Degree of intravascular haemolysis after exposure to 30 × 10^6^
*E. coli* for 2.5 h (All groups *n* = 10). (**E**) The intravascular formation of thrombin–antithrombin complexes (saline *n* = 5, *E. coli* + saline *n* = 10, saline + MRS2500 (25.9 μg/h) *n* = 7, *E. coli* + MRS2500 (25.9 μg/h) *n* = 10). The data are given as single observations and mean ± S.E.M, * *p* < 0.05 and ** *p* < 0.01.

**Figure 3 ijms-21-05652-f003:**
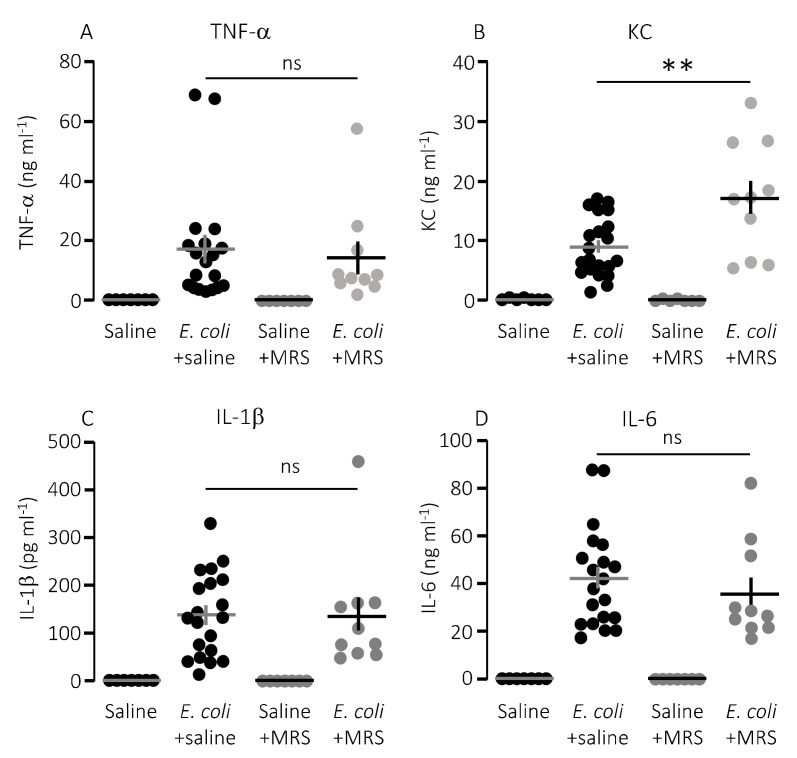
P2Y_1_ receptor antagonist MRS2500 (MRS) increase the proinflammatory cytokine keratinocyte chemoattractant in mice exposed to sepsis with *E. coli*. The level of the following proinflammatory cytokines were measured in plasma from mice exposed to saline or MRS2500 infusion (25.9 µg/h) in the absence or presence of 30 × 10^6^
*E. coli* iv: (**A**) TNF-α; (**B**) keratinocyte chemoattractant (KC); (**C**) IL-1β; and (**D**) IL-6. The data are given as single observations and mean ± S.E.M (saline *n* = 7, *E. coli* + saline *n* = 20, Saline + MRS2500 *n* = 7, *E. coli* + MRS2500 (25.9 μg/h) *n* = 10), ** *p* < 0.01.

**Figure 4 ijms-21-05652-f004:**
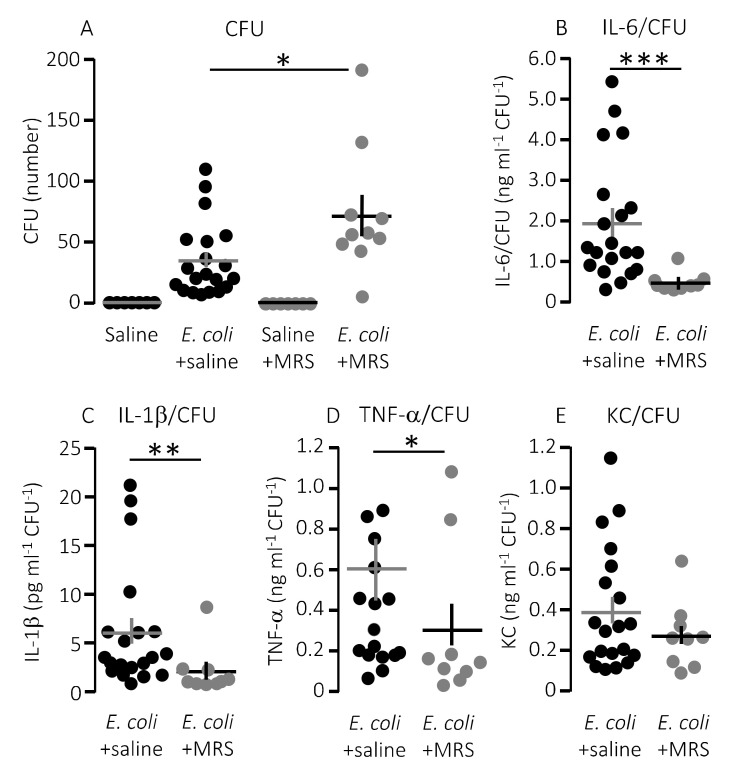
The cytokine response relative to the degree of infection inflicted by injection of HlyA-producing *E. coli* in the absence or presence of infusion of P2Y_1_ receptor antagonist. (**A**) The number of colony forming units (CFU) from streaking out 5 µL whole blood (diluted as described in methods) of mice infused with either control solution or MRS2500 (25.9 μg/h) in the presence or absence of sepsis with HlyA-producing *E. coli*. (B–E) The cytokine [(**B**) IL-6; (**C**) IL-1β; (**D**) TNF-α; and (**E**) KC] response relative to the infection status represented by the number of colony-forming units. The data are given as single observations and mean ± S.E.M (saline *n* = 7, *E. coli* + saline *n* = 20, saline + MRS2500 *n* = 7, *E. coli* + MRS2500 (25.9 μg/h) *n* = 10), * *p* < 0.05, ** *p* < 0.01, *** *p* < 0.001.

**Figure 5 ijms-21-05652-f005:**
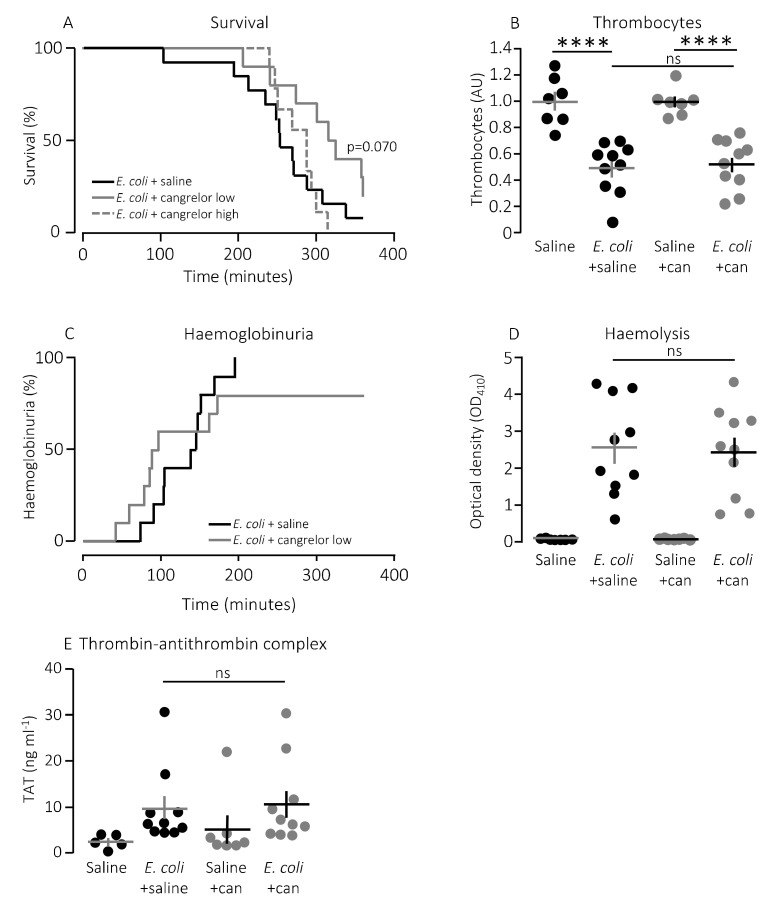
The effect of P2Y_12_ receptor antagonist cangrelor (can) in mice with sepsis. (**A**) Mice with bacteraemia receiving continuous infusion of either saline (*n* = 13) or cangrelor (8.6 μg/h, *n* = 10 or 86.0 μg/h, *n* = 9) for up to 6 h. The mice subjected to bacteraemia received 30 x 10^6^
*E. coli* (WAM1824) iv. Data are shown as Kaplan–Meyer plot. (**B**) The relative number of thrombocytes 2.5 h after injection of 30 × 10^6^
*E. coli* iv (saline *n* = 7, *E. coli* + saline *n* = 10, saline + cangrelor (8.6 μg/h) *n* = 7, *E. coli* + cangrelor (8.6 μg/h) *n* = 10) (**C**) development of haematuria in mice exposed to 30 × 10^6^
*E. coli* for up to 6 h and infused with either saline (*n* = 13) or the P2Y_12_ receptor antagonist cangrelor (8.6 μg/h, *n* = 10) for op to 6 h. (**D**) Degree of intravascular haemolysis after exposure to 30 × 10^6^
*E. coli* for 2.5 h (saline *n* = 7, *E. coli* + saline *n* = 10, saline + cangrelor (8.6 μg/h) *n* = 7, *E. coli* + cangrelor (8.6 μg/h) *n* = 10). (**E**) The intravascular formation of thrombin–antithrombin complexes (saline *n* = 7, *E. coli* + saline *n* = 10, saline + cangrelor (8.6 μg/h) *n* = 7, *E. coli* + cangrelor (8.6 μg/h) *n* = 10). The data are given as single observations and mean ± S.E.M, **** *p* < 0.0001.

**Figure 6 ijms-21-05652-f006:**
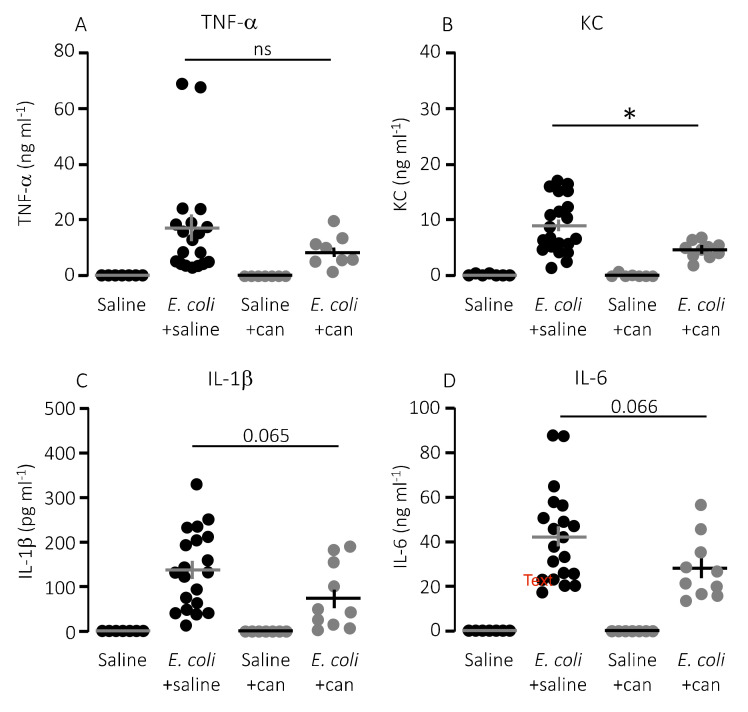
P2Y_12_ receptor antagonist cangrelor (can) does significantly reduce keratinocyte chemoattractant (KC) in mice exposed to *E. coli* iv. The level of the following proinflammatory cytokines were measured in plasma from mice exposed to saline or cangrelor infusion (8.6 μg/h) in the absence or presence of 30 × 10^6^
*E. coli* iv: (**A**) TNF-α; (**B**) KC; (**C**) IL-1β; and (**D**) IL-6. The data are given as single observations and mean ± S.E.M (saline *n* = 7, *E. coli* + saline *n* = 20, saline + cangrelor *n* = 7, *E. coli* + cangrelor (8.6 μg/h) *n* = 10), * *p* < 0.05.

**Figure 7 ijms-21-05652-f007:**
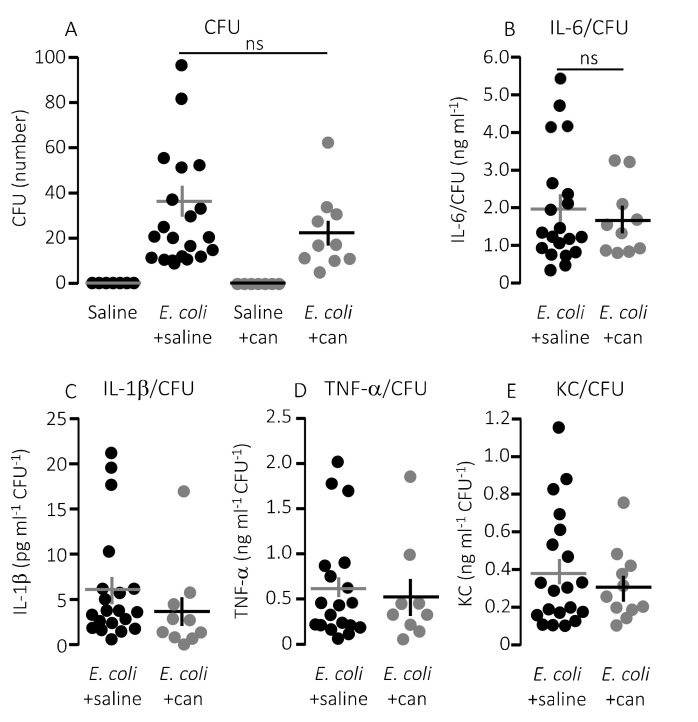
The cytokine-response relative to the degree of infection inflicted by injection of HlyA-producing *E. coli* in the absence or presence of infusion of P2Y_12_ receptor antagonist. (**A**) The number of colony forming units (CFU) from streaking out 5 µL whole blood of mice infused with control solution or cangrelor (8.6 μg/h) either after exposure to HlyA-producing *E. coli* or saline solution. (**B**–**E**) The cytokine [(**B**) IL-6; (**C**) IL-1β; (**D**) TNF-α; and (**E**) KC] response relative to the infection status represented by the number of CFU. The data are given as single observations and mean ± S.E.M (saline *n* = 7, *E. coli* + saline *n* = 20, saline + cangrelor *n* = 7, *E. coli* + cangrelor (8.6 μg/h) *n* = 10.

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
