# Peer review of "Prevention of P2 Receptor-Dependent Thrombocyte Activation by Pore-Forming Bacterial Toxins Improves Outcome in A Murine Model of Urosepsis"

_ijms, 2020, doi:10.3390/ijms21165652_

Round 1

Reviewer 1 Report

In this study, authors tested the involvement of purinergic P2Y1 and P2Y12 receptors in thrombocyte activation by pore-forming bacterial toxins in a mouse model of urosepsis. This work is next in a series of publications from the same group which has established a niche in the research in mechanisms of urinary/renal pathologies. The premise of the study is original, the utilized mouse model of urosepsis is interesting and involves injections of a modified LPS that expresses a single virulent factor, alpha Haemolysin (Hly), which is one of the strong clinical factors during urosepsis. Overall, the paper is clearly written, methods are described in sufficient detail, results are interesting. However, the manuscript needs improvement in the areas of the hypothesis development, presentation of the results. Authors also need to address number of other concerns.

Comments

  1. Authors decided to target P2Y1 and P2Y12 receptors ideally in thrombocytes with the hope to achieve an increase in the survival of mice from sepsis. However, it is not immediately clear why thrombocyte inhibition would be beneficial and moreover how does it lead to better survival. Certainly, thrombopenia is a consequence of sepsis, however whether it is detrimental for survival is not clear. This point needs to be explained better especially when setting up the basis for the hypothesis. This will help strengthen the hypothesis.
  2. Similarly, basic information regarding which exact P2 receptors are expressed in thrombocytes is not presented well. Authors point out that P2X1 receptors are not important to target but it’s not clear to the readers why specifically target P2X1 or P2Y1 and P2Y12 receptors.
  3. As a mechanism for P2Y receptor involvement in the septic process, authors focused on the release of pro-inflammatory markers. To me, it is not totally clear why. Yes, some P2 receptors have strong contributions to the inflammatory response which is also induced by sepsis, however there is no strong basis for P2Y receptor role in that. That may also be the reason for the lack/marginal effects on cytokines. To be more convincing, more supportive information needs to be presented. Related to the same, a conclusion was made that P2Y12 antagonist reduced the levels of pro-inflammatory cytokines, however based on CFU normalized data this doesn’t look like the case (Fig. 7). Also, what would an alternative mechanism for P2Y1 involvement in alpha-Haemolysin (Hly)-mediated thrombocyte activation be?
  4. It is stated that erythrocytes may contribute to Hly-induced processes. What about other cells, like immune cells in the blood? In regard to erythrocytes, the statement on 298-300 is pretty strong as it is not supported by findings in this study. Need to soften and refer to the earlier work.
  5. I wonder if authors tried to analyze the data as median instead of mean values. It can better reflect the data based on the way the spread looks.
  6. Figure 1 panels are not presented in the sequence mentioned in the Results. Which panel on Fig. 1 does the statement on lines 127-129 refer to? Fig. 1D shows Fibronectin and not P-selectin (line 131).
  7. Why not include ADP in the PRP assay, Fig. 1E? Also, the panel is not completely explained in Results.
  8. An interesting observation that MRS increased E. Coli CFU. What is the explanation for this effect? Authors tried to link it to NETs, which is completely clear as NETs are induced to mask bacteria and destroy them.
  9. It is not clear why authors tested the 2 antagonists together while there was no effect of P2Y12 antagonist. Also, there is no explanation for the adverse effects of the combined drugs. Some potential explanations can lie in the pharmacokinetic properties, drug-drug interaction, etc.
  10. Statistical comparison of LtxA+MRS vs LtxA+MRS+PBS is missing. Also, what is the significance between ADP+PBS vs ADP+MRS data?
  11. What was the method of administration of the antagonists?
  12. The description of thrombocyte activation in vitro assay is hard to follow. Please, revise to simplify.

Minor (numbers are line numbers)

  1. There is no need to reiterate the findings at the end of the Introduction. Rather, the development of stronger hypothesis is needed. And a mention how what testing of the hypothesis was achieved.
  2. Some terms need to be presented more consistently. For example, P-selectin or p-selectin, etc.
  3. 235-236 – “dampen the proinflammatory response” – may need to rephrase as there is not much support for this
  4. 58 – logical; 280 – as a virulent factor; “dampen” used a lot; etc. In some places language can be streamlined.

Author Response

We would very much like to thank the reviewers for taking time to evaluate our work and for all the constructive criticism. It has been very helpful.

We have included the point-by-point response in a word file to be able to include supplementary experiments for your consideration.

Reviewer 2 Report

The authors have done a wonderful job.  I especially commend them for their thorough experimental design and careful, measured interpretation of their data. The implications of their work are particularly resonant in the context of Covid-19 disease.  New strategies to control cytokine storm and related outcomes are more valuable than ever. 

Author Response

Thank you very much for the encouraging evaluation. We hope that the current revision will not disturb the overall impression.

Reviewer 3 Report

The authors build on their previously extensive work with alpha haemolysins, in particular from E.coli, and the interaction with P2 receptor signalling in sepsis. 14/57 citations in the reference list provide evidence of the authors significant contribution to the field. The current manuscript addresses the interaction between human or mouse platelets (incorrectly referred to as thrombocytes throughout the manuscript) and bacterial haemolysins. The authors demonstrate that platelet P2Y (ADP) receptors contribute to platelet activation by bacterial haemolysins in vitro and that inhibition of these receptors in vivo improves survival post E. coli sepsis.

There is a major discrepancy between the in-vitro and in vivo experiments. The experiments with human platelets and P2Y antagonists are performed with pore forming toxins from A. actinomycetemcomitans while the animal model is performed with E. coli that produces a distinct calcium dependent pore forming toxin. The results of the in vitro experiments cannot therefore be directly compared to the results of the in vivo experiments. The authors state that E coli HlyA cannot be tested in vitro since it requires addition of calcium which prohibits in vitro experiments due to calcium mediated aggregate formation. This is likely due to calcium induced fibrin clot formation and can be counteracted by addition of a small peptide antagonist of fibrin polymerization (GPRP) as is standard procedure when investigating platelet activation in response to thrombin. The authors should provide evidence that E coli HlyA mediates P2 receptor dependent platelet activation in vitro.

E coli HlyA is not shown to mediate P2 receptor dependent platelet activation in vivo as is implied in the manuscript title. In order to investigate this platelet activation should be monitored in the septic animals with and without P2 receptor blockade either using flow cytometry or detection of platelet activation markers in plasma. This will determine whether platelet activation has occurred and in order to determine if HlyA is responsible for platelet activation an E. coli HlyA mutant should be used for comparative studies.

The circulating platelet count is not a measure of platelet activation in sepsis. It may occur due to platelet activation either directly to bacteria (products) or indirectly due to activation of coagulation, endotheium. It may also be mediated by decreased platelet production or increased clearance. In Figure 2B the platelet counts for E. coli +saline are compared to saline + MRS. This is not the correct comparison to test. It is E. coli +saline and E. coli + MRS that should be compared for significant difference.

The authors show that MRS2500 prolongs survival at 6 hours in Figure 2 but it is difficult to reconcile this prolonged survival with the increased bacteria load observed in the same animals. The adjustment of cytokine levels relative to CFU in Figure 4 implies that the cytokine storm is relatively decreased in MRS2500 treated animals, however this is not conclusive since CFU alone is not the only driver of cytokine production. PAMPs (other than LPS) from dead bacteria or DAMPs from damaged host cells also contribute.

Collectively, while an increased survival is observed in MRS2500 treated mice, this is not matched in cangrelor treated mice and as such the paper lacks mechanistic insight on how survival is improved and whether this is in fact a platelet or pore forming toxin dependent event.

Author Response

We would very much like to thank you for taking the time to evaluate our work and for all the constructive criticism. It has been very helpful. 

We have included the point by point response as a word document to be able to include some supplementary material for your consideration.
